# Heterogeneous Electrocatalysis of Carbon Dioxide to Methane

Yugang Wu [ID], Huitong Du, Peiwen Li, Xiangyang Zhang, Yanbo Yin and Wenlei Zhu *[ID]

State Key Laboratory of Pollution Control and Resource Reuse, State Key Laboratory of Analytical Chemistry for Life Science, the Frontiers Science Center for Critical Earth Material Cycling, School of the Environment, School of Chemistry and Chemical Engineering, Nanjing University, Nanjing 210023, China
* Correspondence: wenleizhu@nju.edu.cn

**Abstract:** Electrocatalytic $CO_2$ reduction to valued products is a promising way to mitigate the greenhouse effect, as this reaction makes use of the excess $CO_2$ in the atmosphere and at the same time forms valued fuels to partially fulfill the energy demand for human beings. Among these valued products, methane is considered a high-value product with a high energy density. This review systematically summarizes the recently studied reaction mechanisms for $CO_2$ electroreduction to $CH_4$. It guides us in designing effective electrocatalysts with an improved electrocatalytic performance. In addition, we briefly summarize the recent progress on $CO_2$ electroreduction into $CH_4$ from the instructive catalyst design, including catalyst structure engineering and catalyst component engineering, and then briefly discuss the electrolyte effect. Furthermore, we also provide a simplified techno-economic analysis of this technology. These summaries are helpful for beginners to rapidly master the contents related to the electroreduction of carbon dioxide to methane and also help to promote the further development of this field.

**Keywords:** carbon dioxide reduction; methane; electrocatalysts; techno-economic analysis

## 1. Introduction

Anthropogenic activities, particularly the extensive utilization of fossil fuels, have caused a significant increase in atmospheric $CO_2$ concentrations, exacerbating the ongoing climate change crisis. Increased levels of $CO_2$ lead to a heightened greenhouse effect, resulting in several adverse impacts on the global climate, including an increase in temperatures and sea level, altered precipitation patterns, and an increased frequency and intensity of natural disasters. These phenomena, in turn, have far-reaching ecological, social, and economic consequences, such as habitat destruction, species extinction, displacement of populations, and loss of biodiversity. Therefore, it is imperative to adopt sustainable energy sources and practices that reduce our dependence on fossil fuels to mitigate the detrimental effects of climate change and ensure a sustainable future for the planet and its inhabitants [1–7]. Thus, the reduction of $CO_2$ to carbon-containing fuels is a promising technology for reducing $CO_2$ emissions and achieving a sustainable future. This approach allows the conversion of intermittent renewable energy into high-energy fuels, providing a pathway to reduce our reliance on fossil fuels. Additionally, integrating $CO_2$ into the global energy cycle through hydrocarbon synthesis allows us to achieve true global carbon neutrality [8–13]. At present, the main technologies aimed at reducing $CO_2$ emissions include photo-, electro-, bio-, thermal, and their synergistic catalyses [14–19]. Each of these methods has its own set of advantages and limitations. For instance, photocatalysis is easy to perform and has a broad range of applications; however, it suffers from poor catalyst stability and lifespan [20]. Biocatalysis involves using biological enzymes to catalyze $CO_2$ in mild-reaction conditions with good selectivity; however, yields are often low and catalyst deactivation is a common issue [21]. In this context, we focus on electrocatalytic technology due to its rapid reaction rate, excellent selectivity, and established industrial infrastructure. Additionally, the proportion of electricity generated by renewable energy sources is

increasingly significant in the total electricity mix [22]. Additionally, the process offers a means of energy storage, making it a viable option for balancing the fluctuating supply and demand of energy. However, the development of cost-effective, large-scale $CO_2RR$ systems is crucial for the successful deployment of this technology. Further research and technological advancements in this area are necessary to advance toward a carbon-neutral future [5,23–29].

Methane, which is among the products of $CO_2RR$, is regarded as a high-value commodity due to its high energy density of 55.5 MJ/kg. Moreover, the methane produced by the electrochemical reduction of $CO_2$ is not emitted into the atmosphere and contributes to the greenhouse effect as the well-established infrastructure for gas pipelines, allowing for the seamless storage, consumption, and distribution of methane, rendering it a widely utilized component of natural gas. With a composition of 21.4% of total primary energy, methane boasts a high abundance and is an attractive candidate for various energy applications [30–34]. Concomitantly, contemporary technology offers the potential to convert $CH_4$ into fundamental chemicals through various routes, such as the oxidative conversion into syngas or direct conversion into other chemical compounds [35–38]. Even after the inevitable transition to thermonuclear energy in the distant future, methane remains the most portable, easily stored, and transported fuel and general-purpose chemical raw material [39]. Most importantly, the next generation of rocket fuel will be liquid methane; the in situ production of methane as rocket fuel on alien planets, such as Mars, will become a key technology for human interstellar navigation. Therefore, the development of $CO_2RR$ technology to prepare high amounts of $CH_4$ is necessary for this application.

However, it is unfavorable to convert $CO_2$ into $CH_4$ due to the presence of $\pi$ bonds in $CO_2$ molecules. In addition to the limitation of sluggish reaction kinetics, the hydrogen evolution reaction (HER) will also compete with $CO_2RR$ [40]. To solve the abovementioned challenges, developing electrode catalysts with a high efficiency and good selectivity is necessary and urgent. In recent years, numerous efforts have been devoted to the design and synthesis of suitable heterogeneous electrocatalysts for the electrocatalytic reduction of $CO_2$, and some of them have shown outstanding $CH_4$ selectivity [41]. To a certain extent, some of the problems have been solved; however, there are still challenges to be faced concerning the $CO_2RR$ to $CH_4$ formation process. Therefore, understanding the reaction mechanism and a summary of the related work based on the $CO_2RR$ to $CH_4$ formation process are necessary, which will contribute to further developments in the field.

Although methane has the advantages of its high calorific value and well-established transportation infrastructure, the market price of methane is relatively low because the development of the technology to exploit shale gas and methane hydrate result in a large global methane supply. Hence, it is imperative to perform a technical–economic analysis of the industrial implementation of $CO_2$ electrolysis concerning $CH_4$ formation. Techno-economic analysis is a fundamental tool for assessing the economic benefits and costs of emerging technologies in practical applications. An assessment of the technical and economic aspects of a novel technology is an indispensable step in the process of translating it from the research into practical application [42–45]. In the context of the electrocatalytic reduction of $CO_2$ for the formation of $CH_4$, a comprehensive economic analysis can provide valuable guidance to researchers for developing catalysts that are more responsive to the complex and dynamic demands of the market.

In this review, we summarize the latest progress in $CO_2RR$ for the selective electrocatalytic reduction of $CO_2$ to $CH_4$ in an aqueous solution based on heterogeneous catalysts. Firstly, we discuss the reaction mechanisms and electrolyte effect, fundamentally, which provides an insight into designing electrolyzers and electrocatalysts with an improved performance. Then, we focus on several electrocatalysts with an excellent catalytic performance and great development potential. A prevalent characteristic shared among the majority of these electrocatalysts is the presence of the copper element, which is expounded upon in the section on the reaction mechanism. In addition, we provide a simplified techno-economic analysis for this technology. Finally, we anticipate electrochemical catalysts'

further development prospects and challenges. We are confident that this review will be helpful for beginners in this field and will further advance the development of $CO_2RR$.

## 2. Reaction Mechanism

Many research efforts have attempted to discover the mechanism of the electrochemical reduction of $CO_2$ to $CH_4$ from both experimental [46–49] and computational points of view [49–54]. From the pure-thermodynamics point of view, it is possible to reduce carbon dioxide to $CH_4$ at a potential of + 0.17 V vs. reversible hydrogen electrode (RHE) [55] (Table 1). However, numerous studies have shown that $CO_2$ electrochemical reduction to $CH_4$ consists of multiple elementary steps [47,52]. As shown in Figure 1A, $CO_2$ is firstly absorbed on the catalyst and hydrogenated into *COOH via an electron transfer–proton coupling process [51]. Then, the *COOH is further evolved into *CO, which is the main branch point to determine whether or not to produce oxygen-containing products. In path I, *CO goes through a CHO* intermediate, with the overall path proceeding as:

$$CO_2^* \rightarrow COOH^* \rightarrow CO^* \rightarrow CHO^* \rightarrow CH_2O^* \rightarrow CH_3O^* \rightarrow CH_4 + O^* \text{ or } CH_3OH^*$$

(the $H^+ + e^-$ reactants and $H_2O$ product formed were left off).

The step to determine the selectivity is the final $CH_3O^*$ reduction step. It was found that the production of $CH_4$ had a more favorable reaction-free energy. In path II, *CO goes through a COH* intermediate with the overall path proceeding as:

$$CO^* \rightarrow COH^* \rightarrow C^* \rightarrow CH^* \rightarrow CH_2^* \rightarrow CH_3^* \rightarrow CH_4^*$$

(the $H^+ + e^-$ reactants and $H_2O$ products formed were left off).

Notably, from Peterson et al.'s work, we know that the absorption energy of the intermediate is crucial for product distributions [56]. For example, the metals (Au, Ag, and Zn) with weak *CO-bound energy produce little methane because CO experiences priority desorption before any further reductions occur [57–59]. Additionally, metals (Pt, Pd, and Ni) with strong *CO-bound energy cannot remove the *CO from the surface because of the highly unfavorable thermodynamic conditions. Thus, $CO_2$ can only be reduced further to $CH_4$ with an exceedingly low Faraday efficiency (FE) on these electrodes [60]. In contrast, the metal Cu is located near the top of the volcano curve of the limit potential. This means that the *CO-adsorption intensity of Cu is suitable for $CH_4$ production via the $CO_2RR$ process (Figure 1B–D).

**Table 1.** Half reactions and potentials of $CO_2$ electrochemical-reduction reactions.

| Half-Reactions Formula | Electrode Potential/V (vs. RHE) |
|---|---|
| $CO_2 + H_2O + 2e^- \rightarrow CO + 2OH^-$ | −0.10 |
| $CO_2 + 2H_2O + 2e^- \rightarrow HCOOH + 2OH^-$ | −0.20 (pH < 4); −0.20 + 0.059 (pH > 4) |
| $CO_2 + 3H_2O + 4e^- \rightarrow HCHO + 4OH^-$ | −0.07 |
| $CO_2 + 5H_2O + 6e^- \rightarrow CH_3OH + 6OH^-$ | 0.02 |
| $CO_2 + 6H_2O + 8e^- \rightarrow CH_4 + 8OH^-$ | 0.17 |
| $2CO_2 + 8H_2O + 12e^- \rightarrow C_2H_4 + 12OH^-$ | 0.08 |
| $2CO_2 + 9H_2O + 12e^- \rightarrow CH_3CH_2OH + 12OH^-$ | 0.09 |

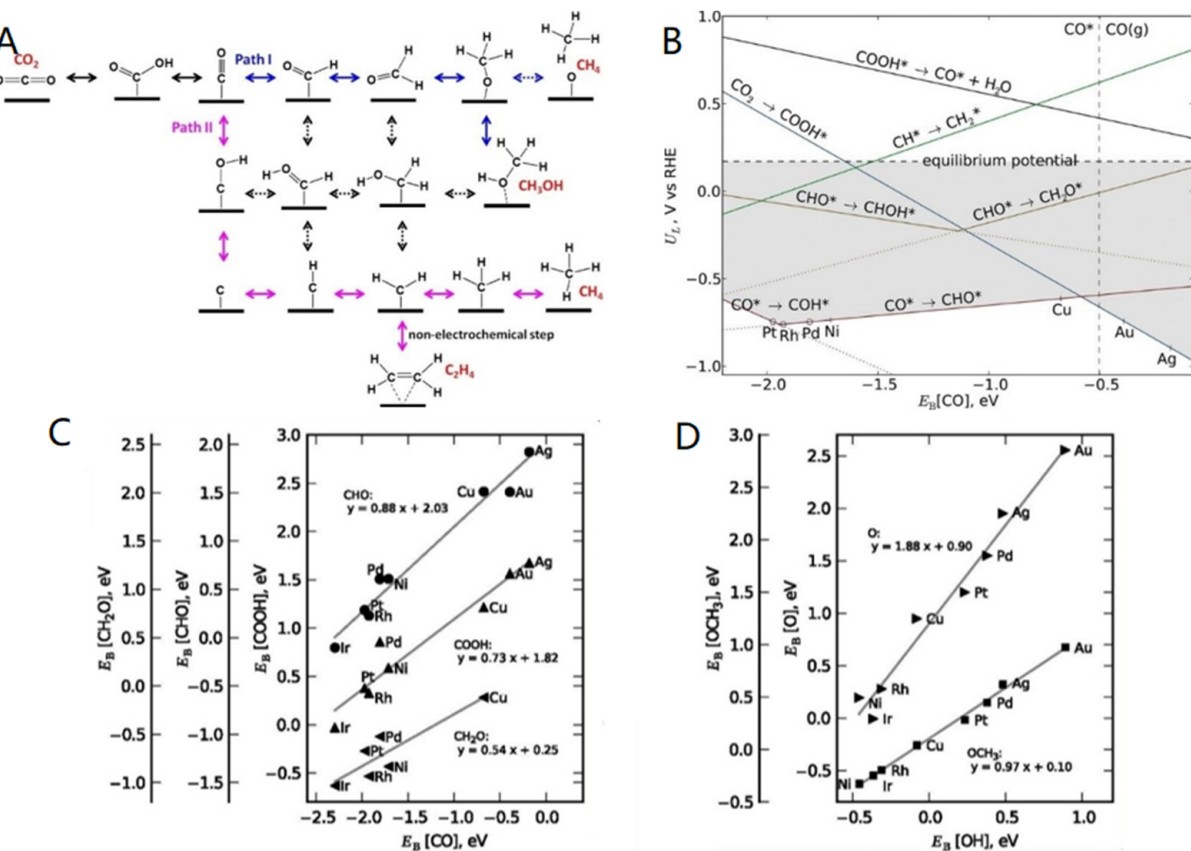

**Figure 1.** (**A**) Pathway of $CO_2RR$ for $CH_4$ formation on Cu (111). Reproduced with permission from Ref. [13]; (**B**) volcano plot of limiting potentials versus CO-binding strength for $CO_2$ reduction; (**C,D**) linear energetic scaling relationships between absorption energy of CO (EB) and certain adsorbed intermediates. Reproduced with permission from Ref. [56].

Notably, some details of the reactions may be slightly different from what was mentioned above over different $CO_2RR$ catalysts. As shown in Figure 2A, Dong et al. [61] reported that *CO protonated through a similar bridge configuration on the $Cu_2O$/Cu interface. This conclusion was confirmed by the density functional theory (DFT) calculation. Interestingly, they also found that the $Cu_2O$/Cu interface formed during the electrochemical reaction process played a crucial role in determining the selectivity of methane formation, which may indicate that the crystal plane is not the key factor for the $CO_2RR$ to $CH_4$ formation process on reconstructed $Cu_2O$ microparticles.

As for the surface-reaction mechanism, two hypotheses were proposed, which are the Eley–Rideal (H comes from the solution) and Langmuir–Hinshelwood (H comes from the surface-adsorbed hydrogen (*H)) mechanisms, respectively (Figure 2B,C). Yogesh and coworkers [62] studied the mechanism of electrochemical $CO_2$ reduced to $CH_4$ on the surface of Cu. They found that the methane production rate was significantly suppressed when increasing the pressure of CO. However, for the Eley–Rideal mechanism, the reaction rate should be positively correlated with the pressure of CO, which was inconsistent with the experimental phenomena. The experimental result thus excludes the Eley–Rideal mechanism and strongly supports the Langmuir–Hinshelwood mechanism, where $CO_{ads}$ and $H_{ads}$ are in competition with each other for surface sites. The result was also confirmed by Asthagiri and coworkers' works with the DFT calculation [51].

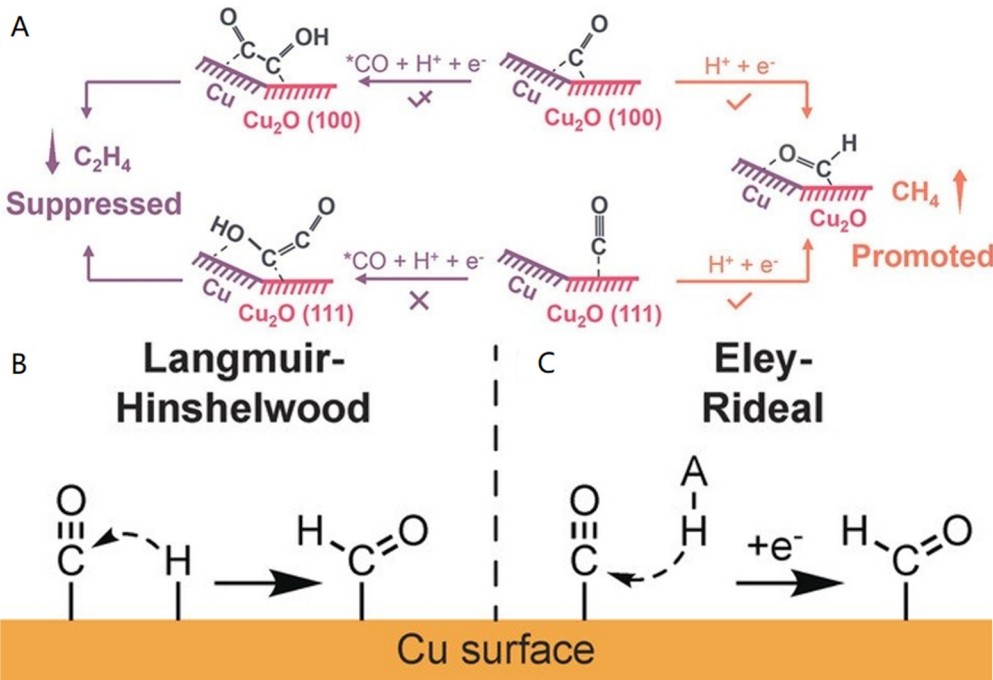

**Figure 2.** (**A**) Reaction pathway and adsorption site of *CO protonation to *CHO, as well as C−C coupling to *OCCOH on reconstructed c–$Cu_2O$/Cu and o-$Cu_2O$/Cu interfaces. Reproduced with permission from Ref. [61]; (**B**) Langmuir–Hinshelwood mechanism; (**C**) Eley–Rideal mechanism. Reproduced with permission from Ref. [62].

## 3. Electrolyte Effect

### 3.1. $CO_2$ Reduction in Aqueous Electrolytes

A mild-reaction condition is usually associated with decreased expenses. Additionally, a lab reactor, such as H-shaped electrochemical and flow cells, is present in an aqueous environment with alkaline electrolytes [63–65]. Thus, understanding the reaction condition in an aqueous environment and the corresponding influencing factors is very helpful.

The environment near the interface of the catalysts, such as the pH and concentration of $CO_2$, is different from bulk electrolytes [66,67]. Therefore, we need an overall understanding of the process to improve the reactivity. First, the pH near the cathode interface greatly impacts the reaction pathways and the formation of certain intermediates. The generated $OH^-$ during the $CO_2RR$ process cannot be immediately transferred to the bulk electrolyte resulting in the pH in the vicinity of the cathode being much higher than that in the bulk electrolyte [68–70]. Ma et al. [71] reported a simple method to determine the local pH experimenting in GDE-based high-rate CO electroreduction. They found that a high local pH facilitated the formation of $C_2$ products. Therefore, we can add buffering agents, such as $KHCO_3$ and phosphate, to the electrolyte to reduce the $C_2$ product and facilitate the formation of $CH_4$. It is well-known that $CO_2$ in water is in acid-base multiequilibrium: $CO_2 + H_2O + OH^- \rightleftharpoons HCO_3^- + H_2O \rightleftharpoons H_2CO_3 + OH^-$. Higher local pH values would decrease the $CO_2$ concentration near the interface of the cathode, resulting in the slow kinetics of $CO_2RR$ [72]. A higher pH also decreases the concentration of the H* intermediate [73]. According to the Langmuir–Hinshelwood mechanism, this will inhibit the formation of $CH_4$. Thus, it is critical to investigate the role of the electrolyte on electrochemical $CO_2RR$.

In addition to the pH effect, hydrated cations can also affect the interfacial interactions occurring at the surface [74–76]. First, hydrated alkali metal cations can serve as a buffer to offset an elevated pH and reduced $CO_2$ concentration in the vicinity of the cathode. The buffering capacity follows the order of $Cs^+ > Rb^+ > K^+ > Na^+ > Li^+$ [77]. According to Chen's group, the intermediates are stabilized by the electric double-layer (EDL) field formed across the Helmholtz layer via the adsorbate dipole-field interaction, which can be

adjusted by changing $M^+$ at the interface [78]. Additionally, Koper et al. [79] found that a $CO_2$ reduction does not occur in the absence of metal cations in the solution. Based on this phenomenon, they proposed that metal cations' main role is stabilizing critical carbon dioxide intermediates. This remarkable observation extends to other common catalysts as well.

*3.2. $CO_2$ Reduction in Non-Aqueous Electrolytes*

Nonaqueous electrolytes can be categorized into three distinct types: ionic liquids (molten salts that are composed of organic cations and organic/inorganic anions), organic liquids (such as acetonitrile, methanol, and dimethyl sulfoxide), and mixed solutions of the two. [80]. Non-aqueous electrolytes usually have a higher $CO_2$ solubility. In methanol electrolytes, the $CO_2$ solubility is five times higher than that in water at room temperature [81]. Additionally, the absence of proton donors in non-aqueous electrolytes creates an environment that depresses the hydrogen evolution reaction (HER) during electrochemical reactions [82]. Furthermore, due to the variety of non-aqueous electrolytes, we can obtain specific products of $CO_2RR$ by modifying the electrolyte [80]. In the realm of non-aqueous electrolytes, the electrochemical reduction of $CO_2$ is commonly believed to follow a series of pathways. Initially, $CO_2$ is activated to create the $CO_2 \bullet -$ anion radical, which is deemed the rate-limiting step. Subsequently, two $CO_2 \bullet -$ radicals dimerize to produce oxalate, or a disproportionation reaction between $CO_2 \bullet -$ and $CO_2$ generates CO and $CO_3^{2-}$. Lastly, in the presence of trace amounts of H2O, $CO_2 \bullet -$ can be protonated to form HCOOH or dissociated to produce CO and $OH^-$ [83].

Despite their numerous advantages, the capital cost of non-aqueous electrolytes is much higher than aqueous electrolytes. Additionally, due to the complex structure of non-aqueous electrolytes, the reaction mechanisms remain poorly understood. Hence, a significant amount of further research is necessary before non-aqueous electrolytes can be effectively implemented in industrial applications [84].

**4. Progress in the Design of Catalysts for $CO_2$ Electroreduction to $CH_4$**

In this section, we presented a range of state-of-the-art catalysts and their corresponding construction strategies in a highly informative manner. In order to facilitate the comprehension and applicability of the presented results, illustrative examples were provided in each section, which serves to provide an intuitive understanding of the catalyst construction process. Additionally, it is noteworthy that certain catalysts displayed exceptional electrocatalytic performances, thus highlighting their potential for further exploration and development. Some of the catalysts and their performers are summarized in Table 2.

**Table 2.** Some catalysts for the heterogeneous electrocatalysis of carbon dioxide to methane.

| Catalyst | Electrolyte | Current Density (mA cm$^{-2}$) | Applied Potential (V) vs. RHE | CH$_4$ FE | Ref. |
|---|---|---|---|---|---|
| FeSA | 1 M KHCO$_3$ | 200 | −1.1 | 64% | [85] |
| Cu-CDS | 0.5 M KHCO$_3$ | 40 | −1.14~−1.64 | 78% | [86] |
| Cu$_{68}$Ag$_{32}$ nanowire | 0.5 M KHCO$_3$ | 80 | −1.17 | 60% | [87] |
| MCH-3 | 1 M KHO | 398.1 | −1.0 | 76.7% | [88] |
| Cu-based cMOF | 1 M KOH | 162.4 | −0.9 | 80% | [89] |
| Zn-MNC | 1 M KHCO$_3$ | 31.8 | −1.8 | 85% | [90] |
| n-Cu/C | 0.1 M NaHCO$_3$ | - | −1.35 | 80% | [91] |
| Cu NW | 0.1 M KHCO$_3$ | - | −1.25 | 55% | [92] |
| Cu−Bi NPs | 0.5 M KHCO$_3$ | 37.2 | −1.2 | 70.6% | [93] |
| NNU-33(H) | 1 M KOH | 391.79 | −0.9 | 82% | [94] |
| Cu$^{2+}$ SA on CeO$_2$ | 1 M KOH | 200 | −0.82 | 65% | [95] |
| CoPc@Zn-N-C | 1 M KOH | 44.3 ± 7.3 | −1.24 | 18.3 ± 1.7% | [96] |
| Ag@Cu$_2$O-6.4 | 1 M KOH | 178 ± 5 | −1.2 | 74 ± 2% | [97] |

### 4.1. Catalyst-Structure Engineering

In heterogeneous catalysis, the catalyst's structure impacts the product distribution of electrocatalyzed $CO_2$ [98–101]. This section focuses on summarizing the different structures of the catalysts, mainly including nanostructured, porous, and single-atom catalysts.

#### 4.1.1. Nanostructured Catalysts

At present, the nano-Cu electrode has been widely studied and used to improve the selectivity and energy efficiency of $CO_2RR$ for $CH_4$ formations [102,103]. The reactivity of $CO_2RR$ for $CH_4$ formations over nanostructured Cu is affected by numerous parameters, such as size, coordinated sites, and morphology [104–106]. When the size of nanoparticles decreases, the radius decreases, which results in the increase in ratio of surface to bulk atoms increasing and a decrease in the average coordination of surface atoms. This phenomenon can also be called the activity–selectivity–size relationship [107]. For example, Peter et al. [108] constructed a series of sizes of Cu nanoparticles (Cu NPs) (diameter: 1.2~20.3 nm) (Figure 3A,B). They found that the catalytic activity and selectivity for $H_2$ and CO products were dramatically increased with the decrease in Cu NP sizes, meaning that the formation of $CH_4$ was inhibited, in particular when the size of Cu NPs was less than 5 nm (Figure 3C,D). In contrast, the bulk Cu catalysts produced $CH_4$ as the primary hydrocarbon product from $CO_2RR$. Buonsanti et al. [109] studied Cu nanocubes (Cu NC) with 24, 44, and 63 nm edge lengths afforded by colloidal chemistry (Figure 3E). As shown in Figure 3F, the cube with a 44 nm edge length has the highest selectivity for $CO_2RR$ at 80%. The surface-atom-density statistical analysis indicated that the edge sites played a key role in the formation of $CO_2RR$. Although Cu NCs did not possess a high selectivity for the $CH_4$ formation, it was observed that the size significantly affected the reactivity of the nanostructured catalysts. As shown in Figure 3G,H, a Cu nanowire (Cu NW) catalyst was reported by Yang et al., and such catalysts exhibit high $CH_4$ selectivity, reaching a $CH_4$ FE of 55% at −1.25 V vs. RHE (Figure 3I–L) [92]. To further study the effect of the morphology of Cu NW on hydrocarbon selectivity, they wrapped the wires with graphene oxide to keep the morphology stable. It was surprising that the selectivity presented no significant change, indicating that hydrocarbon selectivity is sensitive to the morphology of the catalysts.

As the aforementioned nanostructured Cu elements were not supported by any substrate, the particles could easily aggregate during the electrochemical reaction. The nanostructures supported on substrates also received great attention for their superior performance in electrocatalysts [110,111]. As shown in Figure 4A–D, Alivisatos et al. [91] reported a catalyst that Cu nanoparticles supported on glassy carbon (n-Cu/C) capped with tetradecylphosphonate. The catalyst achieved a methanation current density 4 times higher than the pure Cu foil electrode, and its average $CH_4$ FE was 80% during the process of extended electrolysis, which is one of the highest $CH_4$ FE values for room-temperature methanation ever reported (Figure 4E–H). The author proposed that graphene may contribute to lowering the energy barrier of the key step by modifying the electron properties of the anchored Cu nanoparticles due to graphene's unique electronic and physical properties. Additionally, it is easier to increase the Cu–Cu distance on n-Cu/C than that on Cu (111) when the CHO* species is formed on the Cu nanoparticle surface [112]. However, they found that Cu particles supported on glassy carbon can grow during the reaction process, which may be attributed to a combination of particle coalescence and dissolution–redeposition during the electrochemical reaction (Figure 4C,D). The growth of Cu particles impairs the reactivity of the catalyst. Therefore, we need to find strategies to further improve the stability of this catalyst.

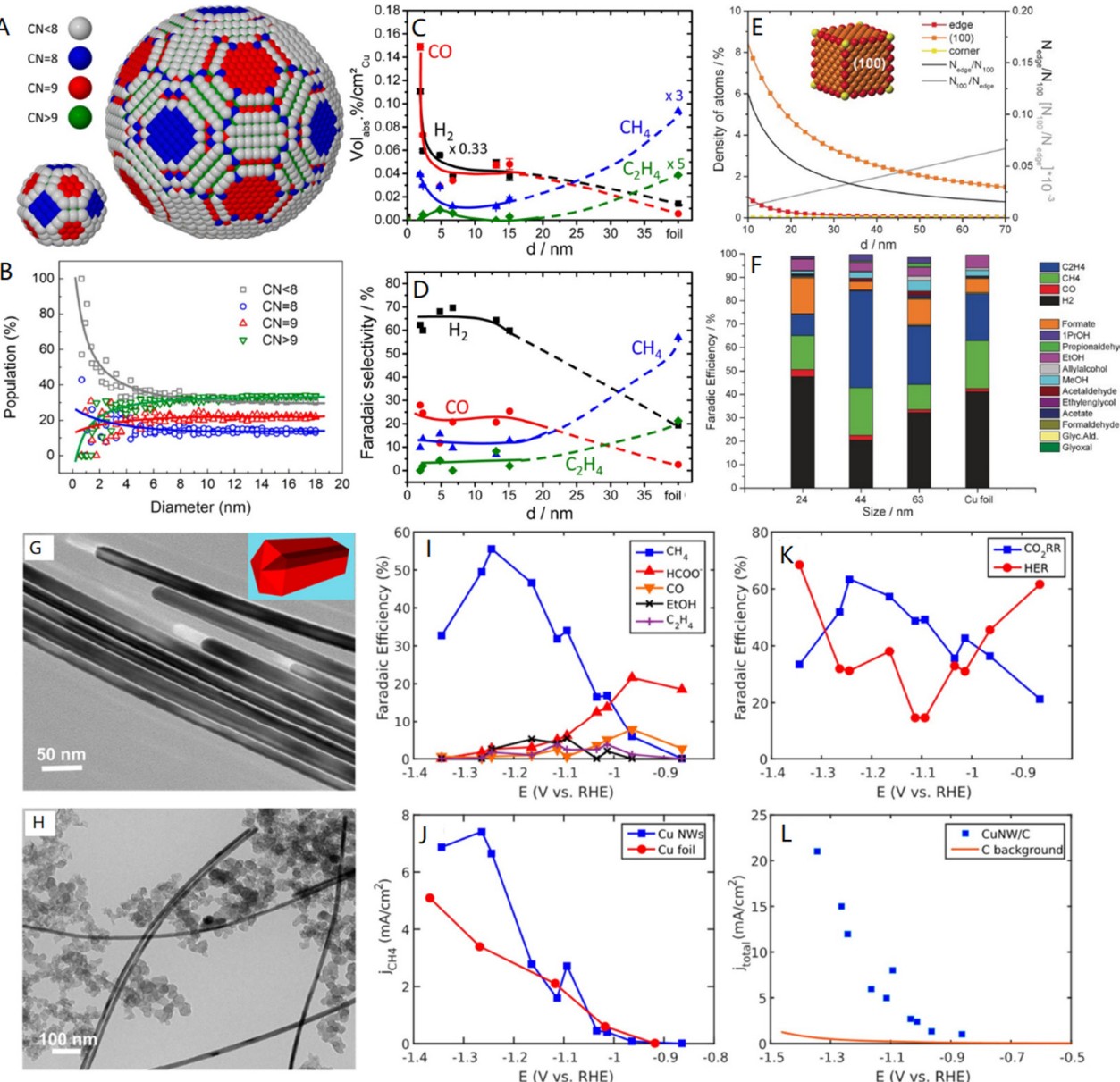

**Figure 3.** (**A**) Models of spherical Cu NPs with 2.2 and 6.9 nm diameters and the quantity of Cu atoms with different coordination numbers; (**B**) population (relative ratio) of surface atoms with a specific CN as a function of particle diameter; (**C**) the composition of gaseous-reaction products; (**D**) FE of reaction products during $CO_2$ electroreduction are a function of the diameter of Cu NPs (reproduced with permission from Ref. [108]); (**E**) density of adsorption sites in Cu NC cubes to the edge length; (**F**) FE of each product in Cu NC cubes and Cu foil at −1.1 V vs. RHE (reproduced with permission from Ref. [109]); (**G**) TEM image of bare wires (the insert shows the 5-fold twinned structure, showing a high proportion of low–coordination edge sites); (**H**) TEM micrograph of Cu NWs loaded on carbon; (**I–L**) Cu NW initial electrocatalytic activity and selectivity. Reproduced with permission from Ref. [92].

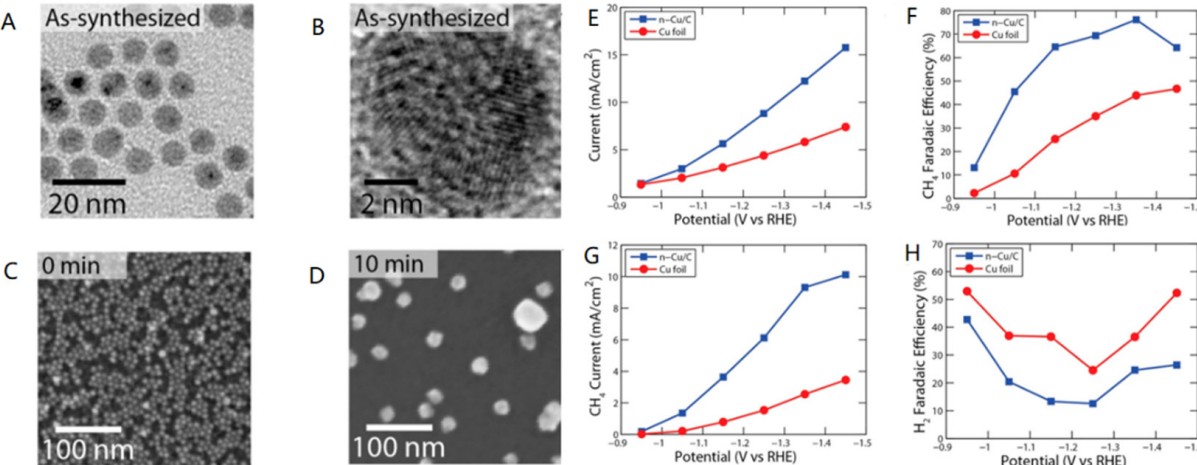

**Figure 4.** TEM images: (**A**) low magnification; (**B**) high-magnification and SEM images of (**C**) as-synthesized and (**D**) evolution of Cu NP supported on glassy carbon after 10 min. (**E**–**H**) Potential (V vs. RHE) dependence on: (**E**) current (mA cm$^{-2}$); (**F**) CH$_4$ FE; (**G**) CH$_4$ current (mA cm$^{-2}$); (**H**) H$_2$ FE of n-Cu/C and Cu foils. Reproduced with permission from Ref. [31].

### 4.1.2. Porous Catalysts

Porous catalysts have attracted considerable attention, recently, because of their large specific surface areas, high density of surface-active sites, efficient mass transfer, and optimization of intrinsic activity [113–116]. In addition, nanopores provide a low coordination position for the reaction [117]. The selectivity of porous catalysts can be changed by increasing the residence time of the intermediates [104]. Because of this characteristic, various porous catalysts for the selective catalysis of CO$_2$ to CH$_4$ and relevant strategies have been developed [118]. Wen et al. [88] reported a perfluorinated covalent triazine framework (FN-CTF-400) that shows an astonishingly selective catalysis of CO$_2$ to CH$_4$ with a dominant competitive advantage over HER. As shown in Figure 5A–F, the CH$_4$ FE value is about 78.7% at potentials between −0.4 and −0.6 V vs. RHE, and what is even more impressive is that the CH$_4$ FE value of FN-CTF 400 can reach 99.3% at the potential between −0.7 and −0.9 V vs. RHE. However, when the potential increases above −1.0 V, the efficiency gradually decreases to 65%. According to the DFT calculations, the high-selectivity depends on the doping fluorine, which regulates the activity of N, making it more conducive to CH$_4$ production (Figure 5G–K). Wen et al.'s outstanding work provides important guidance for designing carbon dioxide electroreduction strategies for more favored materials.

MOF (metal organic framework) and COF (covalent organic framework) are two kinds of crystalline porous materials with a periodic network structure. They have recently been widely used in electrochemistry, especially as an energy-related electro-catalyst for their unique structure. Lan et al. [89] synthesized and studied a series of honeycomb-like porous crystalline hetero-electrocatalysts. This is a core–shell-structured material with HMUiO-66-NH$_2$ as the core (HM stands for honeycomb-like MOF) and COF-366-Cu as the shell (constructed by tetra(p-aminophenyl)porphyrin (Cu-TAPP) and 2,5-dihydroxyterephthalaldehyde (DHA)). MCH-X (X = 1–4) (MCH-X, X = 1–4, X: different MOFs doses in MCH synthesis) was synthesized by adjusting the different amounts of HMUiO-66-NH$_2$ in the COF synthesis system. Among them, MCH-3 presented the best performance with an excellent current density at −398.1 mA cm$^{-2}$ and superior CH$_4$ FE as 76.7% at −1.0V vs. RHE. Rich, open channels of the catalysts facilitated the CO$_2$ adsorption/activation and conversion to CH$_4$ processes. Lan's group also [119] reported a Cu-based conductive metal organic framework (cMOF) that combines electrical conductivity with the porosity of MOF. It is composed of highly conjugated graphene ligands (dibenzo-[g,p]chrysene-2,3,6,7,10,11,14,15-octaol, 8OH-DBC) and Cu ions. Highly conjugated organic ligands endow Cu-DBC with unique redox properties and electrical conductivity (Figure 6A). CH$_4$ FE exhibits up to 80% (Figure 6B,C) accompanied by a

partial current density of +162.4 mA cm$^{-2}$ at a low reduction potential of $-0.9$ V vs. RHE. The abundant and uniformly distributed Cu–O$_4$ sites greatly contributed to the effective ERC-to-CH$_4$ process with high selectivity.

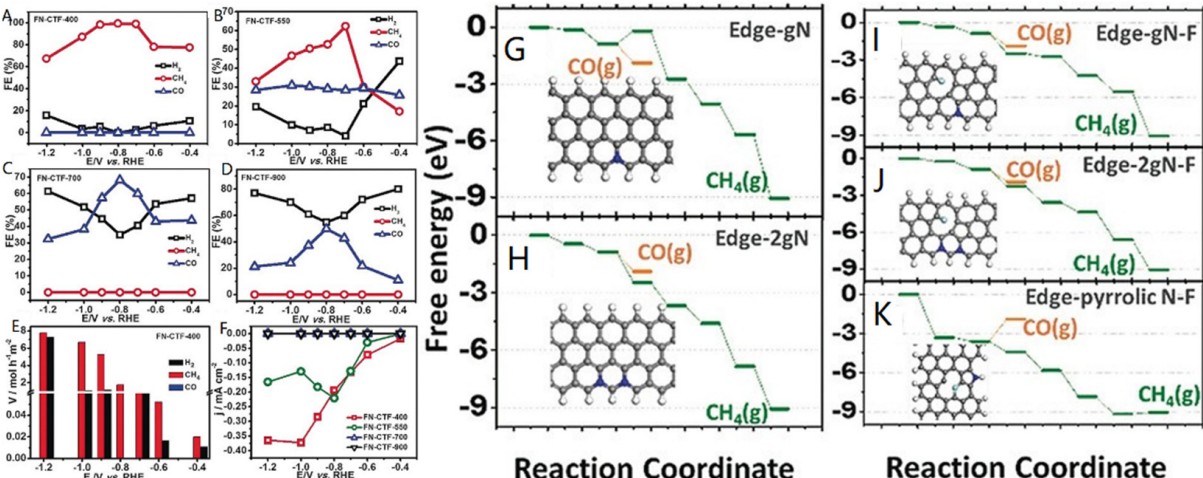

**Figure 5.** Potential of electrodes' dependence on CH$_4$, CO, and H$_2$ FE production: (**A**) FN–CTF–400, (**B**) FN–CTF–550, (**C**) FN–CTF–700, and (**D**) FN–CTF–900; (**E**) hydrogen, carbon monoxide, and methane yields of FN–CTF–400 at different applied potentials; (**F**) corresponds to the current density generated by CH$_4$ on the FN–CTF sample set; (**G**–**K**) reaction models of active sites in N–doped and N– and F–co–doped structures. C: gray, H: white, N: blue, F: cyan. As for the FED, green paths: CO and orange paths: CH$_4$. Reproduced with permission from Ref. [88].

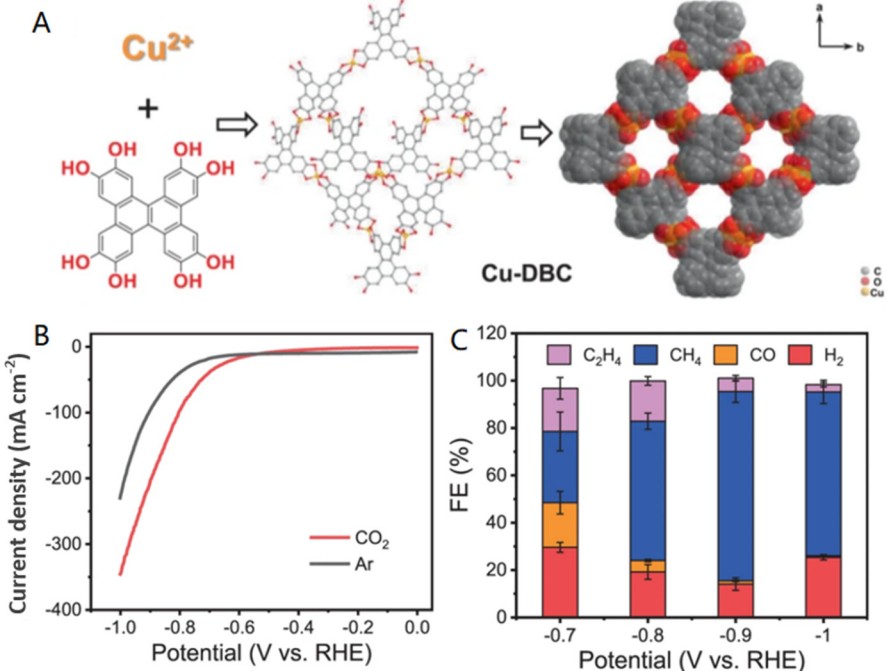

**Figure 6.** (**A**) The structure of Cu-based conductive metal organic framework; (**B**) polarization curves recorded in CO$_2$ and Ar atmospheres; (**C**) FEs of ERC products at different applied potentials. Reproduced with permission from Ref. [39].

Although many achievements have been made in the research and application of porous catalysts, the role of the pore size in the catalytic process is still rarely reported. As the size of the catalysts can affect the mass transfer and density of the activity site, we need to study the role of pore size further.

### 4.1.3. Single-Atom Catalysts

Single-atom catalysts (SACs), in which single metal atoms are anchored to the support, have recently attracted considerable attention [120–123]. The active sites of SACs are isolated metals coordinated by pyridine/pyrrole nitrogen atoms, carbon atoms, or other substrates [124]. The highly isolated active sites of SACs can effectively inhibit the C–C coupling process. Therefore, it can promote the generation of $CH_4$. Additionally, SACs provide the economical, efficient utilization of precious-metal catalysts and open up a broad new field for optimizing the selectivity and activity of various reactions due to their uniform monoatomic dispersions and clear structures [125]. For example, when using single-atom Cu substitute Ce on the $CeO_2$(110) surface, three oxygen vacancies around each Cu site are steadily concentrated, producing efficient carbon dioxide adsorption and activating catalytic centers [126].

Recently, Zhu et al. [86] reported that Cu-embedded carbon dots (Cu-CDS) prepared by calcining $Na_2$ [Cu (EDTA)] $2H_2O$ at 250 °C (the lowest carbonization temperature) converts the carbon-containing molecular complex into solid Cu-CDS, which retains the SAC coordination environment (Figure 7A). The electrocatalytic activity of the catalyst was tested and the results showed that the FE of methane was as high as 78% at the potential of 1.14~1.64 V. Among carbon dioxide-reduction products, 99% were $CH_4$ (Figure 7B–G). The DFT calculations indicate that HER is well-inhibited by $CuN_2O_2$ on the catalyst, which accounts for the high selectivity of $CO_2RR$ for $CH_4$ formation. The easy preparation of this catalyst allows them to have a broader range of application scenarios.

Edward et al. [85] reported a metal–supported monatomic catalytic center. They prepared gas diffusion electrodes (GDEs) by depositing sputtered Cu on a polytetrafluoroethylene (PTFE) substrate and then assembled iron phthalocyanine (FePc) on the Cu surface. By changing the size of the Fe cluster, they found that the affinity of the Fe atom for *CO increased when the size of the Fe cluster decreased. When it decreased to a single site, the affinity for *CO reached the highest point (Figure 8B). As shown in Figure 8C, *CO is transferred to the Fe atom from Cu near the bridge and top sites. The main product of Cu supporting the iron monatomic catalyst was $CH_4$. When the current density was 200 mA cm$^{-2}$, $CH_4$ FE can reach the maximum of 64%, which is much higher than that on bare Cu catalysts with $CH_4$ FE as low as 2%. It may be that the C–C coupling is unfavorable to FeSA compared to the surface of bare Cu; therefore, *CO is more readily hydrogenated to * COH on the Fe site of Cu-FeSA than *CHO when a solvation contribution is present (Figure 8D,E).

Xin et al. [90] reported the electrocatalysis of single Zn atoms supported on N-doped carbon (Zn-MNC) (Figure 9A,B), which was demonstrated by normalized X–ray absorption near-edge structure (XANES) curves, the Fourier transform (FT) k2-weighted extended X-ray absorption fine-structure (EXAFS) spectrum, X–ray absorption spectroscopy (XAS), and X–ray photoelectron spectroscopy (XPS) shown in Figure 9C–F. Compared with the saturated calomel electrode, the catalyst showed a high $CH_4$ FE of 85% with a partial current density of $-31.8$ mA cm$^{-2}$ at a potential of $-1.8$ V. Zn–MNC presented a significant stability improvement since no apparent current drop and great FE fluctuation were observed after 35 h of the electrochemical-reduction reaction (Figure 9G–J). The theoretical calculation shows that a single zinc atom hinders the formation of CO to a large extent, but promotes the formation of $CH_4$. Although the partial current density was low, this proved the feasibility of copper-free elements catalyzing $CO_2$ to hydrocarbons.

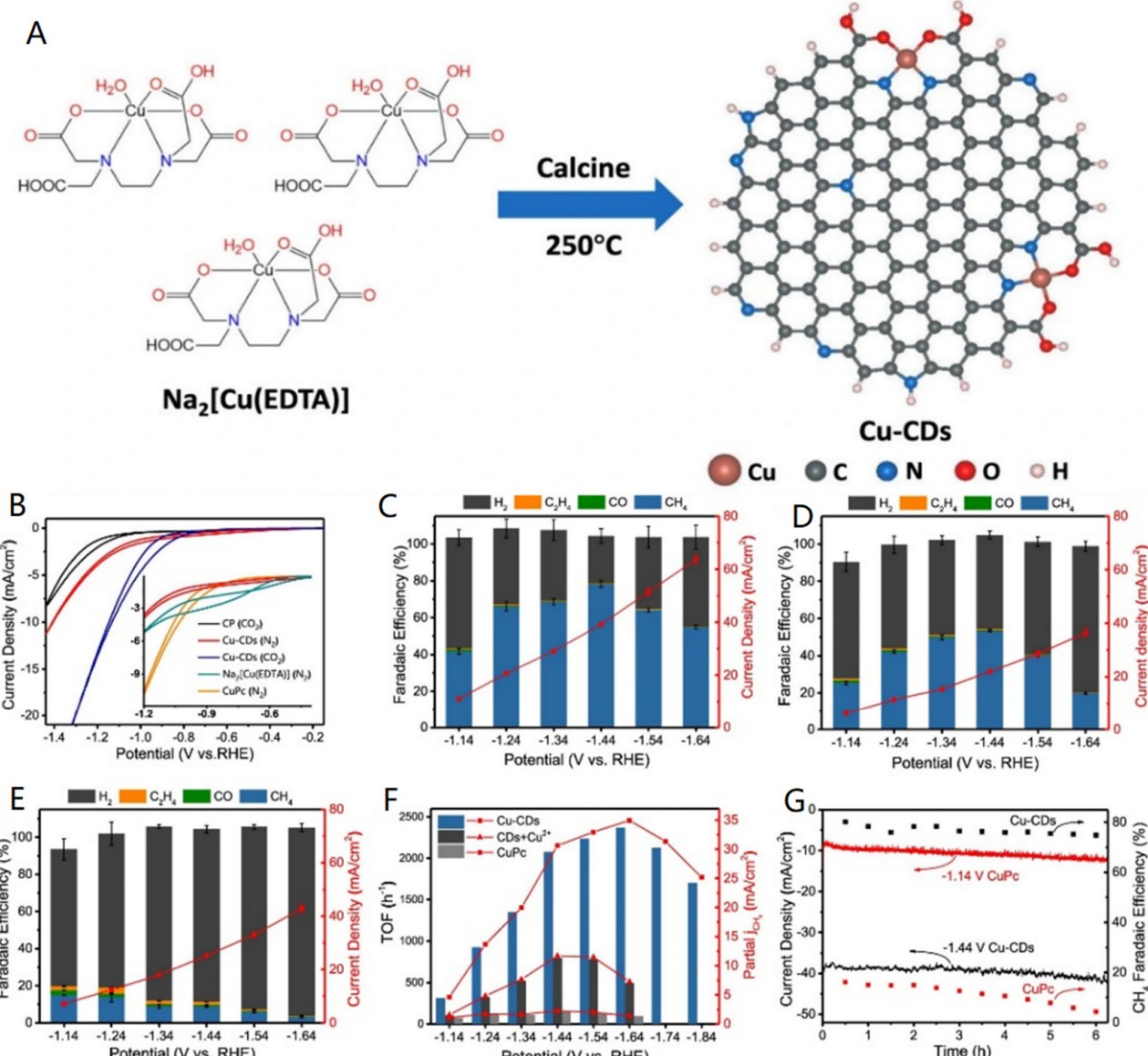

**Figure 7.** (**A**) Scheme of low-temperature roasting process of Cu–CD catalyst; (**B**) CV curves for CP (bare carbon paper), CuPc, $Na_2[Cu (EDTA)]$, and Cu–CDs. The dependence of FE and current density (based on geometric surface area) on the applied potentials of (**C**) Cu–CDs, (**D**) CDs + $Cu^{2+}$, and (**E**) CuPc. (**F**) The $CH_4$ partial current density patterns and TOF spectra of Cu–CDs, CDs + $Cu^{2+}$, and CuPc were studied at different potentials; (**G**) stability tests for Cu–CDs and CuPc at their highest ERC FE potentials. Reproduced with permission from Ref. [86].

To achieve large-scale industrial applications, it is necessary to further improve the stability. Increasing the conversion rate is also an indispensable technique. In addition, if a new type of SAC-preparation method can be developed to improve the preparation process of SACs and simplify their production process, it is also expected to significantly reduce the production cost of SACs [43].

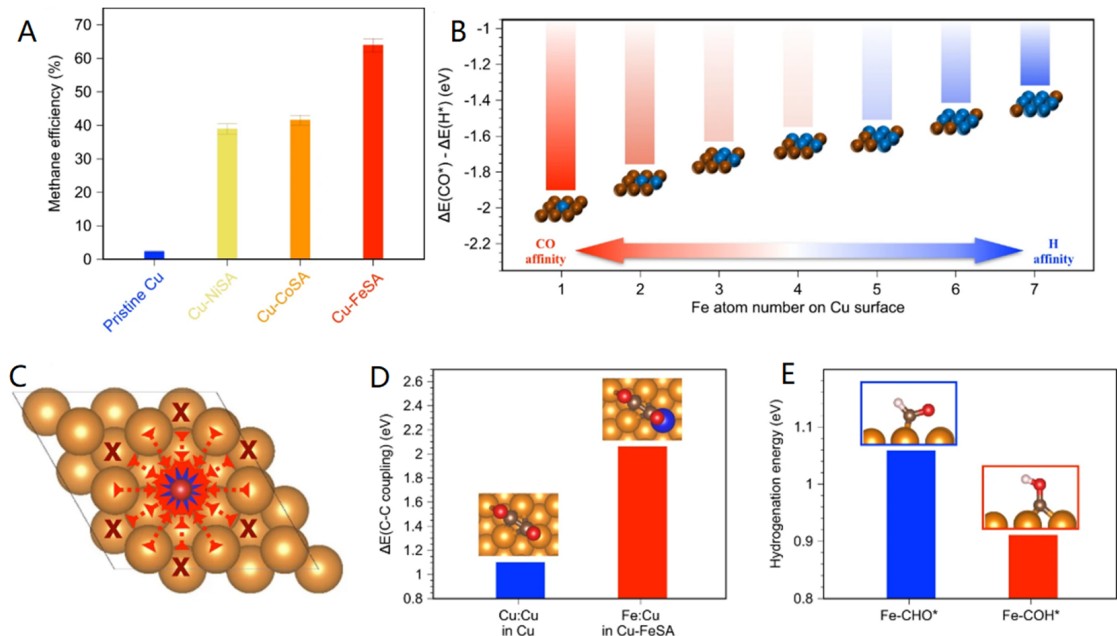

**Figure 8.** (**A**) Catalytic CH$_4$ activities of pristine Cu vs. various single–atom–anchored Cu catalysts for CO$_2$ reduction reaction; (**B**) adsorption energies of *H and *CO are affected by the size of Fe on the Cu surface; (**C**) diagram of * CO transition: the arrow indicates the transition path and the crosshair indicates the fixed * CO–adsorption site; (**D**) coupling energies of pure Cu and Cu–FeSA; (**E**) hydrogenation energy of methanogenic intermediates on Fe center in Cu–FeSA. Reproduced with permission from Ref. [85].

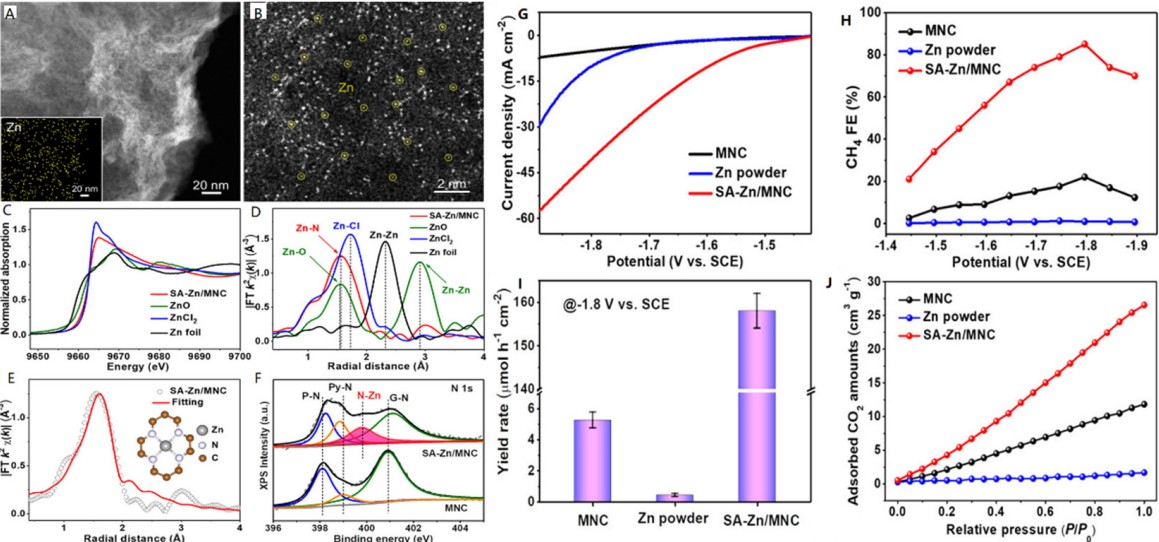

**Figure 9.** (**A**) HAADF–STEM image of an SA–Zn/MNC inset: EDS mapping of Zn. (**B**) Atomic–resolution HAADF-STEM image with some Zn atoms highlighted by yellow circles. (**C**) XANES and (**D**) FT–EXAFS spectra. (**E**) FT-EXAFS. (**F**) XPS. (**G**) Current density, (**H**) FE, (**I**) yield rate, (**J**) and adsorbed CO$_2$ quantities on MNC, Zn power, and SA-Zn/MNC, respectively. Reproduced with permission from Ref. [90].

## 4.2. Catalyst Component Engineering

In addition to the previously summarized strategies for the structural part of the catalyst, we observed that there were also numerous works devoted to tuning the catalyst composition as a way to improve product selectivity. Therefore, this section summarizes the relevant work in terms of alloy, oxidation-state Cu-containing, and tandem.

### 4.2.1. Alloy Catalysts

According to the previous literature, Cu is the only metal that can catalyze $CO_2RR$ to efficient amounts of hydrocarbons and oxygenates due to the suitable adsorption strength of *CO [127,128]. However, it is greatly hindered by poor selectivity and a high overpotential to eliminate the CO from CHO energy barriers on the pure-Cu crystal surface, which is unacceptable for industry-scale applications. To tackle this problem, numerous efforts have been devoted to developing Cu-alloy catalysts [129–131].

Alloying Cu with a foreign metal can improve its electrocatalytic performance, compared to single-metal Cu catalysts, by imparting some unique properties to them, including electronic (changing the electronic structure of the host metal by adding different metals) and geometric (changing the atomic arrangement of actives sites) effect [132–134]. According to the d-band model, the electron effect can change the binding strength of intermediates adsorbed on the surface [135]. Additionally, geometric effects can adjust the binding energy of the intermediates and catalysts, hence tuning their catalytic activities [87]. We can also create bifunctional active centers in which neighboring metals play different catalytic roles, in addition to simply changing the numbers or configurations of specific atoms in the ensemble. The introduction of foreign metals into Cu also changes its surface chemistry, thus changing the distribution of the products [136]. These Cu-alloy catalysts also show significant reactivity behavior to $CO_2RR$ for $CH_4$ formation, outperforming pure metals [130].

Goddard et al. [93] prepared Cu−Bi NPs (Figure 10A–D) through a facile, one-step method, which presented higher activity and selectivity to $CH_4$. The $Cu_7Bi_1$ NPs presented a $CH_4$ FE as high as 70.6% at −1.2 V vs. RHE, which is almost 25 times that of Cu NPs (Figure 10E–J). DFT calculations showed that the addition of bismuth significantly reduced the energy formation of the potential energy-determining step (PDS) for the electrocatalysis of $CO_2$ to $CH_4$. The highly electropositive bismuth absorbed an electron from Cu, causing the Cu to be partially oxidized, which is the active center where $CO_2RR$ is most likely to be converted into $CH_4$.

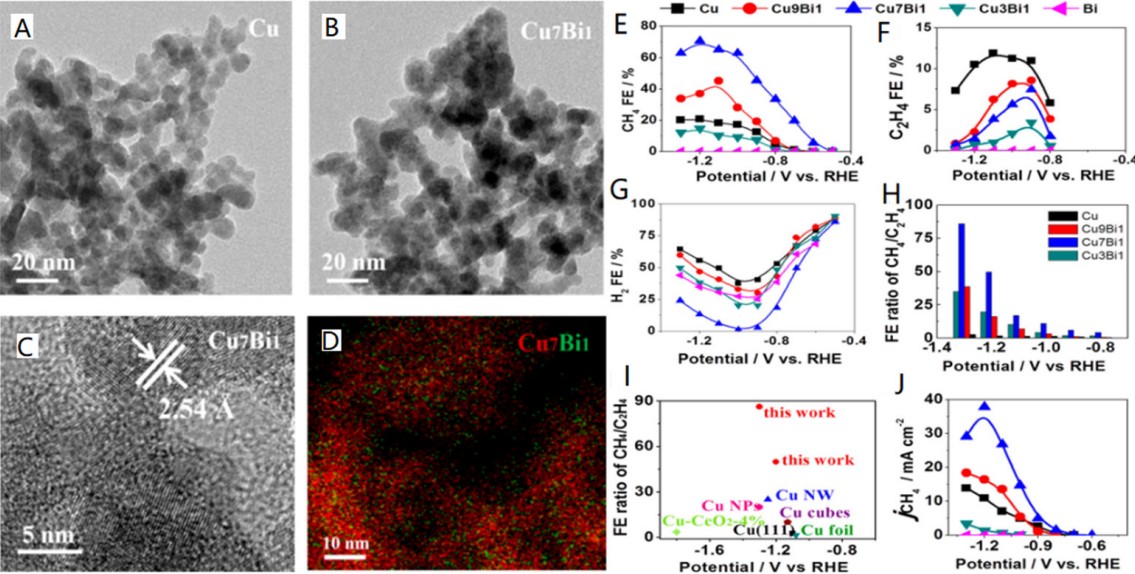

**Figure 10.** TEM images of (**A**) Cu and (**B**) $Cu_7Bi_1$ NPs; (**C**) HRTEM image and (**D**) EDX mapping of $Cu_7Bi_1$ NPs. $CO_2$ electroreduction performance of the synthesized NPs: (**E**) FE of $CH_4$; (**F**) FE of $C_2H_4$; (**G**) FE of $H_2$; (**H**) FE ratio of $CH_4/C_2H_4$; (**I**) comparison of present work with previously reported CH4 selectivity; (**J**) partial current densities of $CH_4$. Reproduced with permission from Ref. [93].

Lee et al. [137] studied a bimetallic Cu/Ag-layered catalyst for eliminating the geometric effect from the electrocatalytic performance by varying the thickness of the Ag layer (Figure 11A). The optimized Cu/Ag–layered catalyst exhibited bifunctional catalytic characteristics that preferentially produced CO (FE = 89.1%) at −0.8 vs. RHE and had a high-selectivity value of $CH_4$ (FE = 65.3%) at −1.2 vs. RHE (Figure 11 B). The silver atoms on the surface of Cu reduced the charge density by forming additional bonds with Cu. With the increase in the thickness of the silver layer, the d–state center gradually shifted down from the Fermi level, which produced weak CO–binding energy on the surface.

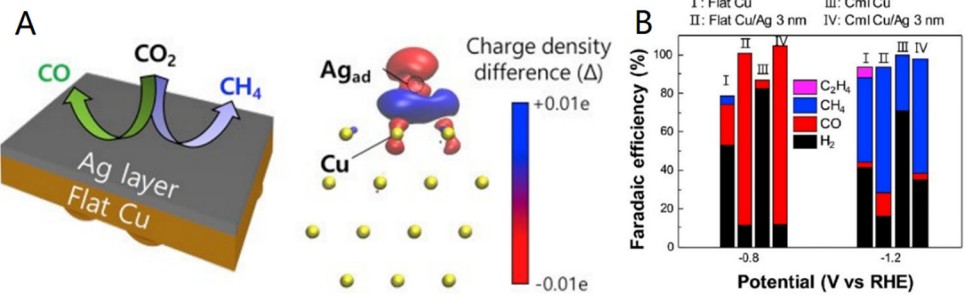

**Figure 11.** (**A**) Cu/Ag-layered catalyst for eliminating the geometric effect from the electrocatalytic performance by varying the thickness of the Ag layer; (**B**) measured FE at −0.8 and −1.2 V vs. RHE. Reproduced with permission from Ref. [137].

### 4.2.2. Oxidation-State Cu-Containing Catalysts

Introducing $Cu^{\&+}$ to the surface of Cu catalysts has been suggested as an active site for $CO_2RR$ [138]. Studies have shown that the stable presence of $Cu^+$ can improve the activity of $CO_2RR$ for $CH_4$ formation [139]. However, in the process of an electrochemical reaction, $Cu^+$ can be easily reduced to Cu due to its instability. Therefore, the oxidation state of Cu may be of great significance in improving its catalytic activity. As shown in Figure 12A, Lan and coworkers [94] synthesized two stable $Cu^+$ coordination polymer (NNU-32 and NNU-33(S) (S = sulfate radical)) catalysts, which showed high selectivity for the electrocatalytic conversion of $CO_2$ to $CH_4$. NNU–33(H) created an impressive $CH_4$ FE amount of 82% at −0.9 V vs. RHE with a partial current of 391 mA $cm^{-2}$, which was one of the best-reported Cu-based catalysts for $CO_2RR$ to produce $CH_4$ (Figure 12B,C). This may account for the greatly enhanced coprophilic interaction observed in NNU-33 (H) and the in situ $OH^-$ substitution of $SO_4^{2-}$ inherent in the molecule, which decreased the Gibbs free energy of PDS (*$H_2COOH$ → *$OCH_2$). The DFT further confirmed this result. The *CO-adsorption energy of Cu-based catalysts increased monotonously with the increase in the oxidation state [136]. Therefore, $Cu^{2+}$ may have a stronger adsorption capacity for *CO. Qiao et al. [95] incorporated $Cu^{2+}$ ions into a $CeO_2$ matrix to obtain stabilizing $Cu^{2+}$ ions. The appearance of $CeO^{2-}$ was demonstrated by in situ Raman spectroscopy, which showed a peak at 560 $cm^{-1}$ originating from the electrochemical reduction of $Ce^{4+}$ to $Ce^{3+}$, indirectly demonstrating the stable presence of $Cu^+$ (Figure 12D,E). The performance was evaluated in the flow reactor for over 6 h, and the average $CH_4$ FE was about 65% at a constant potential of −1.4 V vs. RHE (Figure 12F–I). The DFT calculation demonstrated that stable $Cu^{2+}$ active sites can significantly improve the initial adsorption of CO and promote the hydrogenation of *CO to *$OCH_3$. Both of the abovementioned catalysts showed excellent catalytic performances. It can be seen that maintaining oxidized copper is a good idea for designing catalysts.

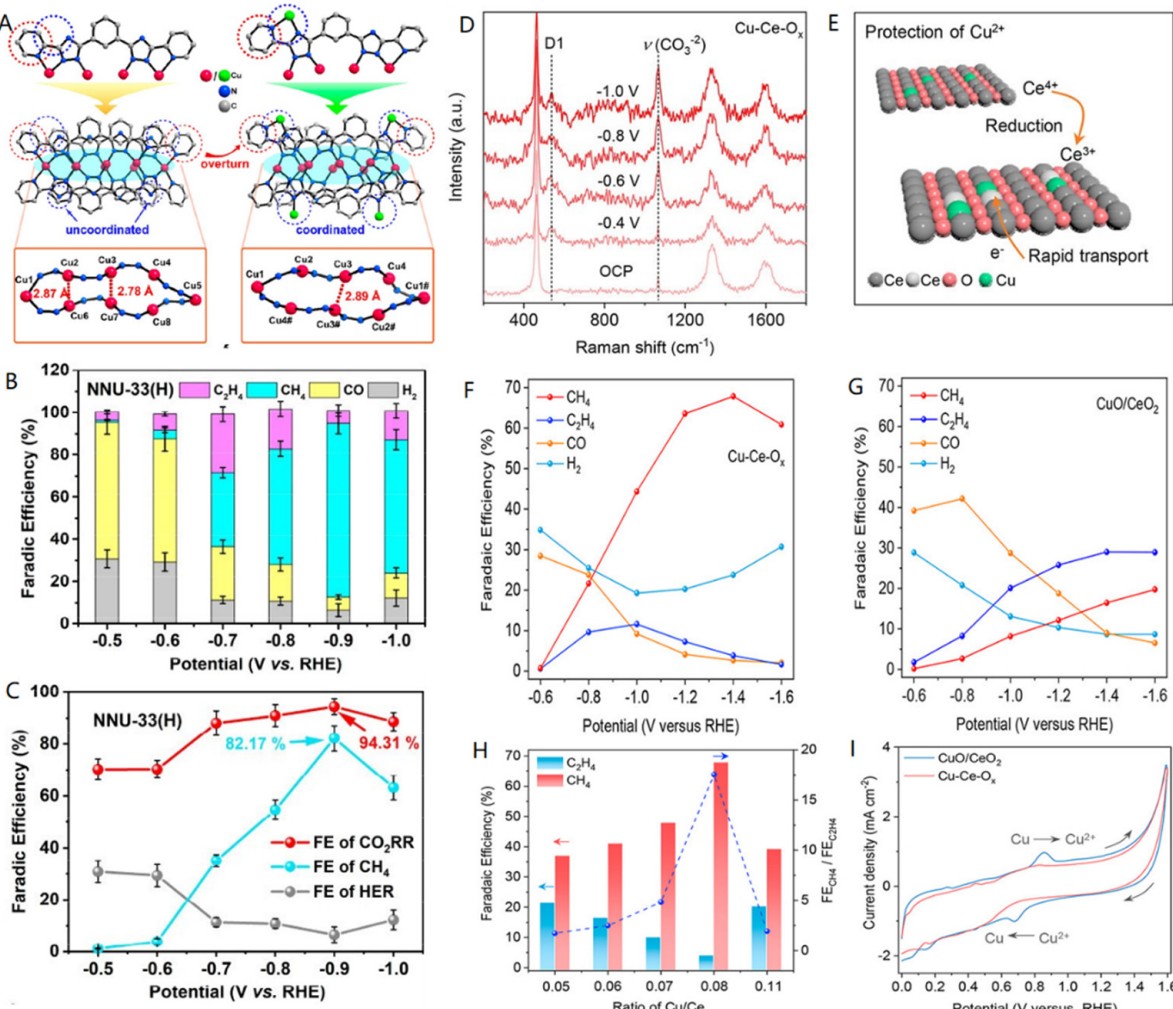

**Figure 12.** (**A**) The crystal structures of NNU–32 and NNU–33(S): NNU-32: coordinated $Cu^+$ ions in NNU–32 (left), NNU–33(S) (right): coordinated $Cu^+$ ions in NNU–33(S); (**B**) NNU–33(H) FE of $H_2$, CO, $CH_4$, and $C_2H_4$; (**C**) comparison of FE to HER, total $CO_2RR$, and $CH_4$ conversion to NNU–33(H); (**D**) in situ Raman spectra for Cu–Ce–Ox collected at different potentials, indicating the existence of $Cu^+$. Reproduced with permission from Ref. [94]; (**E**) self-sacrificing mechanism to protect $Cu^{2+}$; CRR performance of various samples (**F**,**G**); FEs for CRR products at different potentials. (**H**) $FECH_4$ of Cu–Ce–Ox catalysts corrected by IR compared with that of other reported Cu–based catalysts; (**I**) ratio of $FECH_4$ to FE $C_2H_4$ of $CuO/CeO_2$ and Cu–Ce–Ox. Reproduced with permission from Ref. [95].

### 4.2.3. Tandem Catalysts

Cu is one of the only catalysts that can further reduce CO to a more value-added hydrocarbon during the $CO_2RR$. Nevertheless, when CO and CHO are both bound to the same surface, the binding energies follow the liner scaling relationship that limits CO from being reduced further to CHO [56], leading to the disadvantages of high overpotential and low $CH_4$ FE on the single-component Cu catalyst. On the other hand, it is a promising strategy to convert $CO_2$ into CO on more efficient catalysts, such as Au and Ag [140,141], and then reduce the CO generated on Cu to break the limitation of the linear scaling relationship of the key intermediates' adsorption of the abovementioned single Cu catalyst and obtain $CO_2RR$ products with a high selectivity and high yield. Based on this principle, numerous tandem catalysts have been developed, and the key factor to be considered in the design of tandem catalysts is how to efficiently transfer CO intermediates from the catalyst that generates CO to Cu.

Recently, Bao and coworkers [96] reported a Cu-free tandem catalyst consisting of cobalt phthalocyanine (CoPc) and zinc–nitrogen–carbon (Zn–N–C) (CoPc@Zn–N–C) that can effectively and electrochemically reduce $CO_2$ to $CH_4$. $CO_2$ is reduced to CO over CoPc, and the generated CO diffuses to Zn–N–C to convert further into $CH_4$ (Figure 13A,C). Compared with CoPc or Zn–N–C alone, the formation-rate ratios of $CH_4$ and $CO_2$ of this tandem catalyst are over 100 times higher (Figure 13B,D,E).

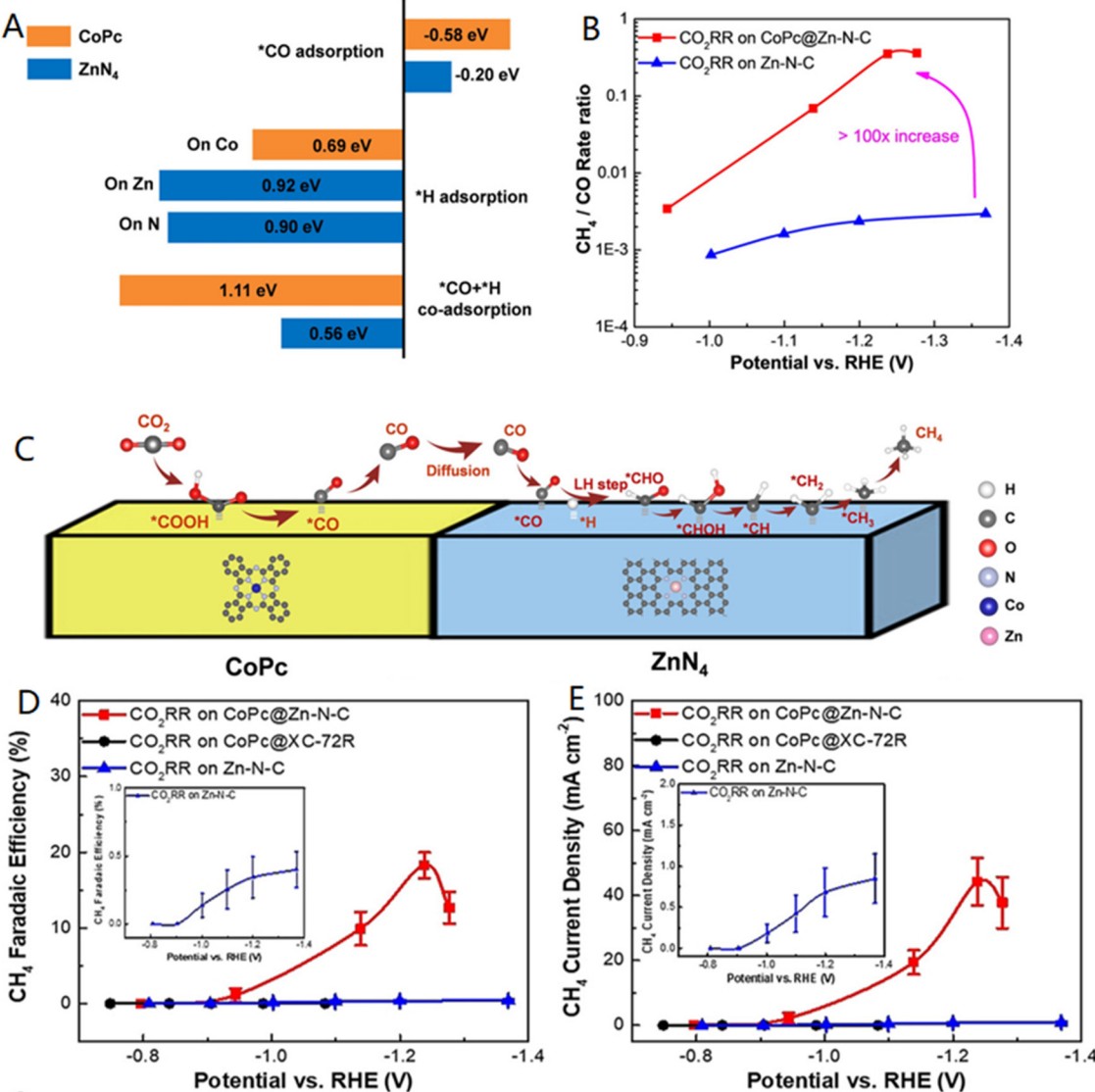

**Figure 13.** (**A**) The adsorption energy profiles of *CO, *H, and the co-adsorption of *CO and H* on CoPc and $ZnN_4$, respectively; (**B**) $CH_4$/CO production rate ratio over CoPc@Zn–N–C and Zn–N–C, respectively; (**C**) reaction mechanism of $CO_2$RR for $CH_4$ formation over CoPc@Zn–N–C; (**D**) $CH_4$ FE; (**E**) potential dependence of $CH_4$ partial current density of $CO_2$RR. Reproduced with permission from Ref. [96].

Peng et al. [97] constructed a yolk-shell nanocell structure comprising an Ag core and a $Cu_2O$ shell that resembled a tandem nanoreactor (Figure 14A–C). Among them, Ag@$Cu_2O$-6.4 NCs (6.4 represents the mole ratio of Cu/Ag) exhibited the greatest $CH_4$ selectivity, achieving a maximum FE value of $74 \pm 2\%$ and a high partial current density of $178 \pm 5$ mA cm$^{-2}$ at $-1.2$ V vs. RHE and $CH_4$ FE as $72 \pm 3\%$ at $-1.3$V vs. RHE with the local current density continuously increased to $214 \pm 9$ mA cm$^{-2}$ (Figure 14D–E). It is worth noting that the performance was almost the best among the most advanced

CO$_2$RR catalysts especially used for CH$_4$ production and met the technical and economic requirements of any commercially feasible CO$_2$RR catalyst with a current density greater than 100 mA cm$^{-2}$. Ag@Cu$_2$O NCs with different Cu$_2$O envelope sizes exhibited different product distributions. This was because varying CO fluxes per unit area at the shell resulted in varying CO coverage on the Cu$_2$O surface, further confirmed by both the experiment and DFT (Figure 14F).

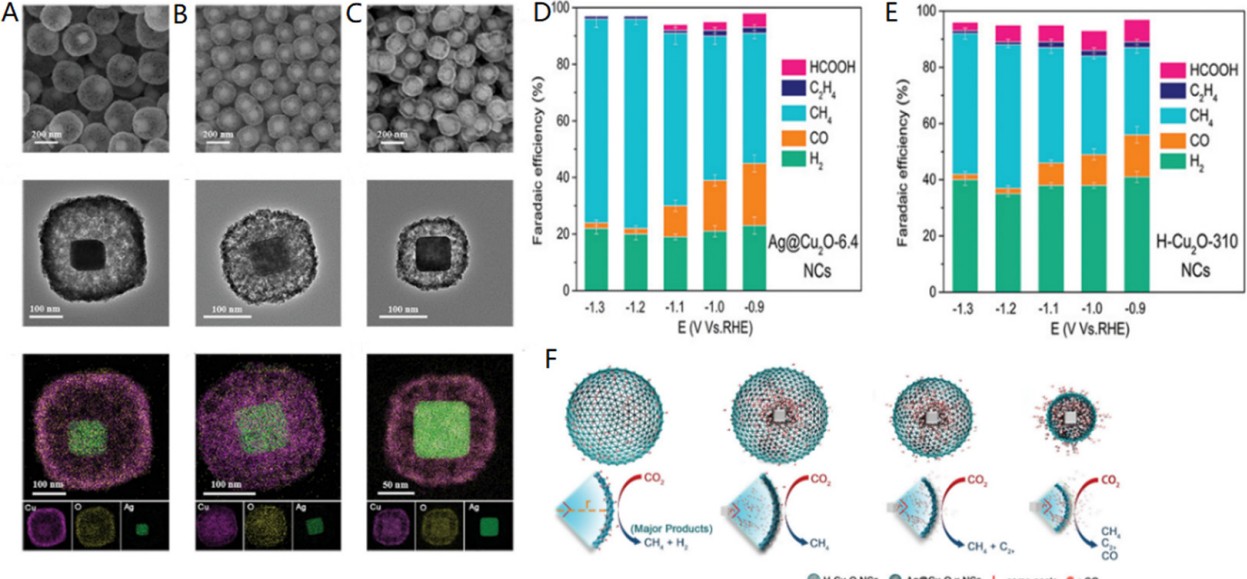

**Figure 14.** Microstructure of Ag@Cu$_2$O–x nanocells: SEM, TEM, and EDX mapping images of (**A**) Ag@Cu$_2$O–6.4 NCs, (**B**) Ag@Cu$_2$O–2.9 NCs, and (**C**) Ag@Cu$_2$O–1.1 NCs. FE of (**D**) Ag@Cu$_2$O–6.4 NCs, (**E**) H–Cu$_2$O–310 NCs; (**F**) schematic diagram shows the CO flux in Ag@Cu$_2$O–x NCs with different outer diameter–Cu envelope sizes and the modulation of the reduced production by the emitted CO molecules from the silver core. Reproduced with permission from Ref. [66].

## 5. Techno-Economic Analysis and Life Cycle Assessment of Electrochemical CO$_2$ Reduction to Methane System

Within this section, a concise examination of the techno-economic analysis pertaining to a general process of the conversion of CO$_2$ to CH$_4$ via electrochemical means is presented encompassing CO$_2$ capture, electrochemical conversion, reactant recycling, and product separation. All the prices used here were based on the Chinese market, to date (the exchange rate of USD to CNY is, at present, 6.88, which will fluctuate over time), and did not take into account the impacts of financial factors, such as carbon taxes or credits. Additionally, due to the absence of commercially developed analogs, a comprehensive analysis of a CO$_2$-reduction process was challenging. Nevertheless, utilizing engineering approximations and making assumptions based on existing technologies can provide valuable insights [142]. We used the net present value (*NPV*) approach to evaluate the feasibility of this technology. The *NPV* was derived through the aggregation of the present values of cash inflows and outflows, which were discounted to the present time using an appropriate discount rate throughout the entire duration of the project or process. If the *NPV* was positive, then the project was considered valuable; if the *NPV* was negative, then the project was considered unprofitable.

$$NPV = \sum_{i=1}^{i=n} \frac{C_i}{(1+r)^i} - C_0$$

where $C_0$ is the initial investment, $C_n$ is the $n$ year cash flow, $i$ is the year, and $r$ is the discount rate.

Figure 15 provides a comprehensive overview of the $CO_2$ to ethylene conversion process. The initial step involved the capture of $CO_2$ from a high partial-pressure stream, such as biogas or industrial flue gas [143–147]. From the information provided by some companies, such as Carbon Clean, the costs of $CO_2$ capture from industrial flue gas through membrane, pressure swing adsorption, and scrubbers were comparable for large-scale processes, ranging between USD 30–$0/ton $CO_2$ [146,148]. However, capturing $CO_2$ from the air is significantly more expensive, with the cost being 5–10 times more than the aforementioned range, rendering it an unviable approach for this study. The capital cost of installing a $CO_2$ capture and storage facility with an annual capacity of 100,000 tons at a steel plant is approximately USD 27 million.

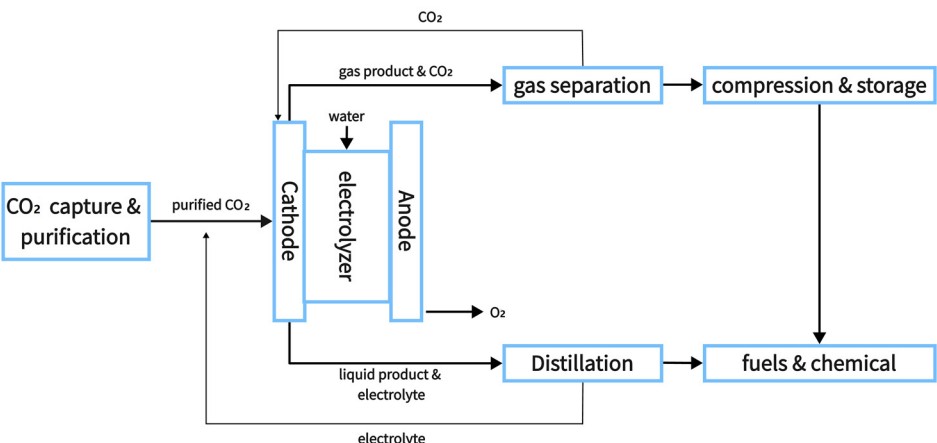

**Figure 15.** A general $CO_2$ electrolysis process.

Here, we coupled carbon capture with electrochemical $CO_2$ convention to cut off the cost of gas transportation [149]. The subsequent step entailed feeding the captured $CO_2$ into a high-pressure (10 bar) GDE-based electrolyzer to produce $CH_4$. It is worth mentioning that the $CO_2$ feed does not necessitate additional pressurization since $CO_2$ derived from biogas plants is often available at high pressures. However, determining the distribution of products was challenging, as it was contingent to various factors, such as temperature, pressure, catalyst type and morphology, cell potential, current density, and pH. Despite the uncertainties, this study assumed a fixed FE of 90% for $CH_4$ and 10% for $H_2$ at $-1.3$ V vs. RHE, respectively.

$$P = I \times V \times t = QV$$

$$Q = It = \frac{M_{CO_2} \times F \times 8}{44 \times FE_{CH_4}}$$

where $P$ is the power required. The total current, $V$, is the cell voltage (here, we did not consider the value of oxygen generated by the anode as a compensation without considering the anode voltage). Therefore, based on Section 4, we set the voltage to $-1.3$ V), $FE_{CH4}$ fixed as 90%. $F$ is the Faraday constant (96,485.334C).

According to the STATE GRID Corporation of China, we knew that the electrovalency was USD 0.091/(kW·h). Therefore, the capital cost of electric power was USD 4.5 million per year. In order to approximate the capital expenses associated with an electrolyzer system, a typical model of an alkaline water electrolyzer stack was utilized. According to several companies that are involved in alkaline water electrolyzers, such as the China Huadian Corporation, the capital cost we obtained for the stack component was USD 300/KW. Therefore, for a capacity of 100,000 tons, the initial stack cost was USD 20 million. Another important factor was that stability pertains to the gradual deterioration or deactivation of the electrode catalyst and the overall electrochemical cell. Here, we established that the electrode material could work for 8000 h per year and the maintenance cost was 2.5%.

The subsequent step was to separate $CH_4$ (account for 0.473), $H_2$ (account for 0.21), and unconventional $CO_2$ (account for 0.315) (as the conversion of $CO_2$ was rarely reported, we set the conversion rate as 60%, which can be achieved by a well-designed electrolyzer). Pressure swing adsorption (PSA), membrane, and low-temperature separations are usually applied to gas product separation, [150–152] But achieving a purity level higher than 99% through membrane separation was challenging. Therefore, we opted to employ pressure swing adsorption (PSA) separation as a means of separating methane and hydrogen [153–155]. Technical details can be found in the ref. [156]. According to Augelletti et al. [156], we can obtain a relatively high concentration of methane gas at a low power cost (270 kJ/kg). The proposed methane fee was intended to specifically target the natural gas and petroleum industries and would entail a cost of USD 300 per ton of methane, which is the lowest price for $CO_2$-reduction products (Table 3). A reference cost of USD 1,990,000 per 1000 $m^3$/h capacity was used [142]. According to the National Energy Administration, the price of hydrogen was USD 5.09 per kilogram.

**Table 3.** Market prices of $CO_2$-reduction products. Ref. [142] notice: The data presented in this table are for 2018.

| Product | Number of Required Electrons | Market Price (USD/kg) | Normalized Price (USD/electron) $\times 10^3$ | Annual Global Production (Mtonne) |
|---|---|---|---|---|
| Carbon monoxide (syngas) | 2 | 0.06 | 0.8 | 150 |
| Carbon monoxide | 2 | 0.6 | 8 | |
| Formic acid | 2 | 0.74 | 16.1 | 0.6 |
| Methanol | 6 | 0.58 | 3.1 | 110 |
| Methane | 8 | 0.18 | 0.4 | 250 |
| Ethylene | 12 | 1.3 | 3 | 140 |
| Ethanol | 12 | 1 | 3.8 | 77 |
| n-Propanol | 18 | 1.43 | 4.8 | 0.2 |

As shown in Table 4, we summarized the capital and operating costs of $CO_2$ electrolyzers. We briefly examined various parameters, including $CO_2$, electricity, and selling prices of the final product, which significantly impacted the cost analysis. The financial model does not incorporate the expenses related to sales, labor, and inflation. We observed that, no matter how we optimized the reaction conditions and reduced the costs, we did not make a profit, as the market price of $CH_4$ was too low and the electrovalency was too high. However, this does not mean that this technology is not desirable, because it is very promising to use methane as an energy-storage medium for when controlled nuclear fusion is improved or almost all electricity is generated from renewable energy and used as next-generation rocket fuel; the in situ production of methane as rocket fuel on alien planets, such as Mars, will become a key technology in human interstellar navigation. Another interesting point is that the FE of $CH_4$ has a minor impact on profitability, as its byproduct, hydrogen, is even more expensive.

**Table 4.** Cash-flow sheet.

| Initial Capital Cost ($C_0$) | | Cash Flow (per Year) | |
|---|---|---|---|
| $CO_2$ capture facility | −USD 27 million | Electricity | −USD 64.5 million |
| Electrolyzer cost | −USD 20 million | Maintenance | −USD 1.49 million |
| PSA facility | −USD 12.7 million | Cell compartment replacement (normalized to each year) | −USD 1.28 million |
| | | $CO_2$ capture | −USD 4 million |
| | | sales of $CH_4$ | +USD 10.908 million |
| | | sales of $H_2$ | +USD 10.3 million |
| total | −USD 59.7 million | total | −USD 50.06 million |

## 6. Conclusions and Outlook

Electrochemical $CO_2$ reduction has gained considerable attention as an effective means of mitigating environmental pressure due to its eco-friendliness, operational simplicity, and economic efficiency. Of particular interest is the potential for enhancing the selectivity of $CH_4$ production in the catalytic process. In this review, we presented an overview of the related research on the catalytic mechanisms and catalyst design strategies, providing an assessment of the state-of-the-art work and techno-economic analysis and life cycle assessment of electrochemical $CO_2$ reduction to methane system, and offering recommendations for future studies.

We briefly described the electrolyte effect to provide a preliminary understanding of the system reactions. With the ongoing research and development, a more thorough understanding of the reaction mechanisms is expected to yield additional strategies for designing high-performance $CO_2RR$ catalysts. A comprehensive mechanistic study, particularly in the reaction pathway catalyzing the multi-electron transfer of $CO_2RR$ for $CH_4$ formation, is essential to improve catalyst selectivity for $CH_4$ products.

Moreover, the development of new powerful toolkits, including machine learning, macrodynamic simulations, and operating conditions/in situ techniques, holds promise for advancing our mechanistic understanding. These tools have the potential to yield insights into the underlying processes that govern catalyst performance, facilitating the development of more efficient and effective catalysts for electrochemical $CO_2$ reductions. Overall, a continued effort in this area of research is essential to address environmental challenges and create a sustainable future. The design and development of catalysts are expected to make significant progress in the future. In this regard, we should make the following efforts in the future:

(1) Improve in situ techniques and apparatus with higher temporal and spatial resolutions to capture key species not previously found experimentally to better understand the reaction mechanism [157,158]. For example, Lu et al. [159] made a breakthrough in the study of the mechanism of the electrocatalytic reduction of $CO_2/CO$ by using advanced techniques, such as electrochemical reaction activity testing and high-pressure in situ spectroscopy. By introducing the strategy of probe molecules acting on the target reaction network, they proposed a new perspective on the surface-coverage level of important intermediates and the $CO_2/CO$-reduction reaction network, which makes up for the cognitive deficiencies, at present, and provides a new idea for development in this field;

(2) Develop high-throughput syntheses and testing techniques for the rapid and reproducible screening of catalysts. High-throughput approaches are particularly suitable for problems where the parameter space is too large to be effectively solved using conventional methods [160–162]. Catalyst synthesis and testing fit this perfectly, and unsurprisingly, it can help the development of $CO_2RR$ electrocatalysts;

(3) Develop accelerated DFT methods and microscopic dynamics for machine learning modeling. This can help us throughly and accurately explain the mechanisms and rapidly predict catalyst materials [163]. Singh et al. [164] developed high-accuracy neural network (NN) ML models for predicting the adsorption energies of COOH*, CO*, and CHO* [165–168]. This work accelerated the development of catalysts and provided an effective strategy to circumvent the scaling relation.

Finally, although the low market price of methane makes it impossible to commercialize electrocatalytic $CO_2RR$ for $CH_4$ formation, we should consider improving the performance of catalytic materials, such as electrolysis voltage, current density, energy efficiency, and stability, as it is very promising to use methane as an energy-storage medium for when controlled nuclear fusion is improved or almost all electricity is generated from renewable energy and used as next-generation rocket fuel, where the in situ production of methane as rocket fuel on alien planets, such as Mars, will become a key technology in human interstellar navigation. In order to promote the industrialization of electrocatalytic carbon dioxide, we should pay more attention to studies on upstream and downstream processing, process design and techno-economic feasibility.

**Author Contributions:** Conceptualization, Y.W. and H.D.; validation, P.L., X.Z. and Y.Y.; formal analysis, H.D.; investigation, P.L.; resources, X.Z.; data curation, Y.Y.; writing—original draft preparation, Y.W.; writing—review and editing, Y.W.; visualization, Y.W.; supervision, W.Z.; project administration, W.Z.; funding acquisition, W.Z. All authors have read and agreed to the published version of the manuscript.

**Funding:** W.Z. would like to acknowledge the support from the National Natural Science Foundation of China (22176086), the Natural Science Foundation of Jiangsu Province (BK20210189), the State Key Laboratory of Pollution Control and Resource Reuse (PCRR-ZZ-202106), the Fundamental Research Funds for the Central Universities (021114380183, 021114380189, 021114380199), the Research Funds from Frontiers Science Center for Critical Earth Material Cycling of Nanjing University, and the Research Funds for Jiangsu Distinguished Professor.

**Institutional Review Board Statement:** Not applicable.

**Informed Consent Statement:** Not applicable.

**Data Availability Statement:** The data presented in this study are openly available in [10.1021/acs.iecr.7b03514], reference number [142].

**Conflicts of Interest:** The authors declare no conflict of interest.

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
