# Peer review of "Heterogeneous Electrocatalysis of Carbon Dioxide to Methane"

_methane, doi:10.3390/methane2020012_

Round 1

Reviewer 1 Report

The authors focused on the Electrocatalytic conversion of CO2 to CH4 using heterogeneous catalysts, especially on Cu-based catalysts. They discussed from the catalyst engineering and electrolyte effect. The fundamental mechanism of the reaction is also covered. In the end, the future outlook in the field is given as well. However, there're some issues in the paper.

1. The justification for CH4 generation is not enough and adequate in the introduction. The authors need to compare it with other potential reduction products in a bigger picture. Some aspects include the market share and market price. 

2. The references are not very new. Most of the cited papers published couple of years ago. I was wondering if the authors can highlight some most recent examples (2020-now). Here is a recommended review paper published this year for the authors attention (DOI: 10.1039/d2nr02894h). Besides this one, other papers need to added so that the readers can get the most up to date information in the field.  

Reviewer 2 Report

This paper summarizes the recent advances in the electrocatalytic reduction of CO2 to CH4. The authors focused on reviewing the Cu-based catalysts. What about other catalysts?

I suggest that the authors summarize the different methods of converting CO2 to CH4, then explain the electroreduction process and its advantages compared to the other strategies.

Can the electroreduction of CO2 to CH4 be performed in other electrolytes other than the aqueous electrolytes? If yes, examples are needed to be included in the review.

The authors claimed that one of the advantages of converting CO2 to CH4 is to mitigated the greenhouse effect. CH4 is also a greenhouse gases and it is even more powerful greenhouse than CO2. More explanation is needed regarding this point.

Round 2

Reviewer 1 Report

The quality of this paper has been improved.